# Safety and Efficacy of a Single Procedure of Extraction and Reimplantation of Infected Cardiovascular Implantable Electronic Device (CIED) in Comparison with Deferral Timing: An Observational Retrospective Multicentric Study

**DOI:** 10.3390/antibiotics12061001

**Published:** 2023-06-02

**Authors:** Carlo Tascini, Simone Giuliano, Vittorio Attanasio, Luca Segreti, Andrea Ripoli, Francesco Sbrana, Sergio Severino, Chiara Sordelli, Sara Hana Weisz, Agnese Zanus-Fortes, Gabriele Maria Leanza, Novella Carannante, Andrea Di Cori, Maria Grazia Bongiorni, Giulio Zucchelli, Stefano De Vivo

**Affiliations:** 1Infectious Diseases Clinic, Department of Medicine (DAME), University of Udine, 33100 Udine, Italy; carlo.tascini@asufc.sanita.fvg.it; 2Infectious Diseases Clinic, Azienda Sanitaria Universitaria del Friuli Centrale (ASUFC), 33100 Udine, Italy; simone.giuliano@asufc.sanita.fvg.it (S.G.); zanusagnese@gmail.com (A.Z.-F.); 3First Division of Infectious Diseases, Cotugno Hospital, Azienda Ospedaliera Dei Colli, 80131 Napoli, Italy; rosa.zampino@tin.it (V.A.); novellacarannante@ospedaledeicolli.it (N.C.); 4Second Division of Cardiology, Cardiac-Thoracic and Vascular Department, University Hospital of Pisa, 56126 Pisa, Italy; l.segreti@ao-pisa.toscana.it (L.S.); a.dicori@ao-pisa.toscana.it (A.D.C.); m.g.bongiorni@med.unipi.it (M.G.B.); g.zucchelli@ao-pisa.toscana.it (G.Z.); 5Bioengineering Department, Fondazione Toscana Gabriele Monasterio, 56124 Pisa, Italy; ripoli@ftgm.it; 6Lipoapheresis Unit, Reference Center for Diagnosis and Treatment of Inherited Dyslipidemias, Fondazione Toscana “Gabriele Monasterio”, Via Moruzzi 1, 56124 Pisa, Italy; ifcfsbrana@ftgm.it; 7UOSD Cardiologia, Cotugno Hospital, Azienda Ospedaliera Dei Colli, 80131 Napoli, Italy; sergioseverino@ospedaledeicolli.it (S.S.); sordellichiara@libero.it (C.S.); sarahanaw@yahoo.it (S.H.W.); 8UOC di Elettrofisiologia, Studio e Terapia delle Aritmie, Monaldi Hospital, 80131 Napoli, Italy; stefano.devivo@tin.it

**Keywords:** CIED infection, single procedure, device extraction, antibiofilm

## Abstract

(1) Background: Infections are among the most frequent and life-threatening complications of cardiovascular implantable electronic device (CIED) implantation. The aim of this study is to compare the outcome and safety of a single-procedure device extraction and contralateral implantation versus the standard-of-care (SoC) two-stage replacement for infected CIEDs. (2) Methods: We retrospectively included 66 patients with CIED infections who were treated at two Italian hospitals. Of the 66 patients enrolled in the study, 27 underwent a single procedure, whereas 39 received SoC treatment. All patients were followed up for 12 months after the procedure. (3) Results: Considering those lost to follow-up, there were no differences in the mortality rates between the two cohorts, with survival rates of 81.5% in the single-procedure group and 84.6% in the SoC group (*p* = 0.075). (4) Conclusions: Single-procedure reimplantation associated with an active antibiofilm therapy may be a feasible and effective therapeutic option in CIED-dependent and frail patients. Further studies are warranted to define the best treatment regimen and strategies to select patients suitable for the single-procedure reimplantation.

## 1. Introduction

CIED implants are lifesaving, albeit not riskless procedures. Rates of implants have increased over the past years due to the interplay of several factors, including increasing age and life expectancy and new pacing indications. Such patients frequently present with significant comorbidities and have an increased susceptibility to infectious complications [1,2,3]. Hence, as more complex patients undergo CIED implants, a concomitant rise in infections has also been observed [4,5,6,7], to the extent that the CIED infection rate is greater than the implants themselves [8]. CIED infections are associated with substantial morbidity and mortality [9] and impact quality of life, healthcare resource use, and costs [2,10]. Recent joint guidelines reported a more than two-fold increase in in-hospital mortality, with an estimated in-hospital 30-day mortality due to CIED infection of 5%–8% [1]. The most common pathogens responsible for CIED infections are Gram-positive bacteria, especially *S. epidermidis* and *S. aureus*, while Gram-negative bacteria and fungi are rarer microorganisms in this kind of infection [3,11,12]. The pathogens responsible for CIED infections are biofilm-forming microorganisms in most cases [13]. This microorganism feature makes CIED infections extremely difficult-to-treat diseases. Guidelines consider mandatory leads extraction, as treatment of CIED infections typically includes the removal/extraction of the entire infected CIED system, debridement, and administration of systemic antibiotics to eradicate the infection [1,14]. However, there is poor and conflicting data concerning the timing of the new device re-implant [1,4,14,15]. As with other device infections, such as prosthetic cardiac valve endocarditis, reimplantation cannot be delayed until infection eradication in case of surgery, and the relapse rate in these cases is very low [16].

Same-day extraction and reimplantation in patients with infected CIEDs is a viable but non-standardized option, with only small cohorts of patients [17,18,19,20] and scarce data on patients with systemic infections, particularly endocarditis.

The aim of this study was to assess the outcome of a single procedure of extraction and contralateral reimplantation in patients with CIED infections.

## 2. Materials and Methods

### 2.1. Definitions

Hospital admission was defined as the date of the patients’ admission to the hospitals included in the study. Hospitalization was defined as the period between the hospital admission and the date of discharge from the hospital. CIED infections were defined according to the European Heart Rhythm Association (EHRA) criteria [1]. Local infection was defined as an infection localized to the CIED pocket. Endocarditis was defined according to the European Society of Cardiology’s 2015 modified criteria [21]. Vegetation was defined as an intracardiac mass on the valve or the implanted intracardiac material, identified with transthoracic and/or transesophageal echocardiography. The infecting pathogens were determined with blood cultures and/or lead cultures at the time of diagnosis or extraction. Anti-Gram-negative antibiotic therapy was defined as an antibiotic therapy with displayed activity against Gram-negative bacteria. Antibiofilm therapy/combination was defined as an antibiotic therapy with activity against the common biofilm-producing bacteria isolated in our cohort (the antibiofilm therapy used in our cohort was daptomycin/rifampicin alone or in combination with a beta-lactam molecule) [22,23,24,25,26].

### 2.2. Population Study

Patients presenting in the centers of Pisa and Cotugno (Naples) hospitals between January and December 2019 requiring pacemaker or defibrillator system extraction and new device implantation were retrospectively assessed. Patients with a local, systemic infection or CIED-related infective endocarditis were enrolled.

Patients treated at the Pisa University hospital, considered a national referral center for challenging CIED infections, underwent standard-of-care two-stage replacement, whereas patients in Naples, Cotugno hospital, were treated by means of one-stage device reimplantation on the same day. The Pisa cohort was considered the reference SoC cohort. Of notice, patients referred to the Pisa center are often considered at higher risk of complexity and negative outcomes; therefore, in the SoC, only pacemaker infections were enrolled because these infections are usually less severe compared with indwelling defibrillator infections.

Patients were followed up at both institutions for up to 12 months post-procedure. Those lost to follow-up were also included in the final analysis.

All patients underwent device extraction and new device implantation. In the single-procedure cohort, the benefit of early reimplantation of a contralateral device in the setting of device dependence was considered to outweigh the small theoretical risk of reinfection of the new device. This was also due to the features of the center, which did not enable a SoC procedure. Clinical and procedural data were collected along with therapeutical intervention long-term outcomes.

### 2.3. Transvenous Lead Extraction Procedure

The method used as the standard of care at Pisa hospital was an evolution of the technique introduced by Byrd in the 1980s [27]. Polypropylene dilating sheaths (Cook Vascular Inc., Leechburg, PA, USA) were used to dilate the adherences between the leads and venous/cardiac structures, usually with a single-sheath technique using a standard stylet instead of a locking stylet. The dilation was attempted when manual gentle traction was unsuccessful. In case of failure from the venous entry side, a jugular-vein or multiple-vein approach was considered to succeed (the so-called Pisa approach). Free-floating lead extractions and combined approaches for challenging leads were performed using an intravascular workstation (Cook Vasc. Inc., Leechburg, PA, USA) with tip-deflecting wires, baskets, and loop retrievers [28]. Additional intravascular tools included catchers (remote-control clippers) and lassos (remote-control loop retrievers) by Osypka Gmbh (Grenzach-Wyhlen, Germany).

Powered sheaths with lasers were used in the Naples approach [29,30,31]. This technique is characterized by using locking stylets and laser energy to overcome the adherences. A laser sheath 12/14/16F with a “cool” pulsed ultraviolet laser at a wavelength of 308 nm was used to remove all the adherences on the leads (Spectranetics Laser Sheath).

All the procedures were performed with aseptic technique; surgical services were available in the departments and emergency cardiovascular surgical staff was on standby at all times.

### 2.4. Microbiology and Medical Treatment

The microbiology of the infections was documented with cultures obtained from the removed leads (proximal and distal parts) and/or infected material. Blood cultures were collected at the discretion of the treating physicians when considered appropriate.

Patients received antimicrobial treatments based on the pathogen or epidemiological features. Treatment regimens were modified according to renal functions and patient characteristics at the discretion of the treating physician. Antibiofilm combinations were generally used to achieve microbiological clearance.

### 2.5. Statistical Analysis

Depending on their distributions, the variables were represented as mean +/− standard deviation, median and inter-quartile interval, or sum and percentage. Accordingly, comparisons between groups were performed with an unpaired sample *t*-test, Mann–Whitney test, or chi-square test with continuity correction. A value of *p* less than 0.05 was considered statistically significant. All the analyses were performed with R statistical software [32].

## 3. Results

Overall, 27 and 39 patients were enrolled from the single-procedure and SoC sites, respectively. All demographic data along with predisposing factors, device type, infection, and treatments are reported in Table 1 and Table 2. No significant differences in demographic characteristics or predisposing factors were detected among the two cohorts, with the exception of weight. All devices/leads were extracted from all patients. The device type differed among the groups: while all SoC devices comprised pacemakers (PMKs), the single-procedure group presented with a variety of devices, including PMKs (40.7%), implantable cardioverter defibrillators (ICDs) 22.3%, and biventricular defibrillator implants (BDIs) 37.0%. No significant differences were observed when the infection diagnoses were assessed, with similar distributions for endocarditis and local infections (66.7% vs. 43.6%, *p* = 0.082 in patients with endocarditis; 33.3% vs. 56.4%, *p* = 0.144 in patients with local infections, for single-procedure versus SoC, respectively). Positive blood cultures for bacteria at hospital admission were reported in 12 (44.4%) and 13 (33.3%) patients from the single-procedure and SoC groups, respectively (*p* = 0.511). Patients with sepsis were reported in 11.1% and 5.1% of cases in the single-procedure and SoC cohorts, respectively, with no significant differences among the groups (*p* = 0.393).

Vegetations were reported in 14 (51.9%) and 15 (38.5%) of patients from the single-procedure and the SoC cohorts, respectively. Vegetation size (median, range-mm) ranged from 3.12 ± 0.91 cm in the single-procedure cohort to 4.07 ± 1.12 cm in the SoC cohort.

The time from device implant to extraction differed among the two cohorts, as well as the time from diagnosis to extraction. In the single-procedure group, the time from implant to extraction ranged between 1 and 7 years, with a median time of 2 years, while in the SoC group there was a significantly longer lag time to extraction, ranging from 4 to 13 years, with a median time of 9 years (*p* = 0.002). Similarly, in the single-procedure cohort, the median interval time from diagnosis of device infection to extraction was significantly shorter (median time, range = 27 (12–30 days) compared with SoC (median, range = 60 (30–90) days from diagnosis to extraction) (*p* < 0.001).

A total of 58 strains were isolated overall, 35 in the single-procedure and 23 in the SoC group. *S. epidermidis* was the most prevalent pathogen (37.1%, in the single-procedure and 30.4% in the SoC group), followed by *S. aureus* (20.0% single-procedure and 26.1% SoC). All pathogen distributions are reported in Figure 1. Methicillin resistance rates varied from 33.3% for *S. haemolyticus* to 100% for the single strain of *S. hominis* in the single-procedure group. Only *S. epidermidis* and *S. aureus* showed methicillin resistance (42.9% and 16.7%, respectively) in the SoC group, compared with 61.5% and 42.9% for the same pathogens, respectively, in the single-procedure group. Other pathogens, including Gram-negative bacteria and fungi, were also isolated. Tissue cultures and catheters sent for culture most often yielded *S. epidermidis*. Refer to Figure 1 for further data.

Overall, 33.3% and 53.6% of patients in the single-procedure and the SoC cohorts, respectively, were undergoing antibiotic therapy upon hospital admission, with no differences in the two sets of patients. Anti-Gram-negative antibiotic therapy was reported in similar proportions (25.9% and 23.0% in the single-procedure and SoC groups, respectively). Upon hospital admission, 88.9% of patients from the single-procedure group and 100% of patients in the SoC group were administered antibiotics. Significantly different proportions of patients received an active antibiofilm antimicrobial therapy (including daptomycin or rifampicin, alone or in combination with a beta-lactam, especially novel fifth-generation molecules). Active antibiofilm therapy was administered in 81.5% of the single-procedure patients, while only 38.5% of patients from the SoC group received similar treatments, with a significant difference between the two groups (*p* = 0.001). Anti-Gram-negative antibiotic therapy was administered in 74.1% of the single-procedure and 38.5% of SoC cases (*p* = 0.006), respectively. Data concerning treatments and devices are detailed in Table 1.

Overall, 18.5% and 12.5% of patients were lost to follow-up (FU) in the single-procedure and SoC groups, respectively, while the rest were followed up for 12 months post-procedure. No serious adverse events related to the procedure or the antibiotic therapy occurred. No re-infection was reported in the followed-up population.

Considering those lost to FU, there were no statistically significant differences in mortality rates among both cohorts, with survival rates of patients available at follow-up at one year of 81.5% in the single-procedure group and 84.6%% in the SoC group (*p* = 0.737).

## 4. Discussion

As rates of CIED implants increase, the concomitant growing complexity of patients and procedures implies greater infection rates, consequently hindering life expectancy and quality.

Guidelines and consensus suggest waiting for the absence of signs and symptoms of infections (local and/or systemic) and for persistently negative blood cultures for at least 72 h post-extraction. The waiting period should be extended to 14 days before reimplanting in the presence of valvular vegetations. Evidence and indications are supported by the low quality of the evidence and expert consensus [1,14,33], and there are currently no randomized trials addressing the timing of reimplantation in CIED infections.

As a result, the currently advised timing from explant to implant is still based on limited evidence; therefore, the higher risk of infection in early reimplanting may be due to inadequate infection eradication. On the other hand, delaying reimplantation may be just as threatening due to potential adverse events related to the absence of the devices or electrical therapies, especially in heart failure patients, along with the potential acquisition of nosocomial infections [34]. Of course, the need for a temporary pacing lead in pacemaker-dependent patients waiting for reimplantation can increase the risk of thrombosis and perforation. Bridging with leadless devices is nowadays available to reduce this issue, though poor evidence is available on the possible benefits [35]. Hence, decisions on proper treatment should consider the overall factors and risks and should be customized according to feasibility and patient features. While waiting a couple of days may be reasonable in some instances presenting with lower severity, a 2- to 3-day delay while waiting for cultures to clear before device reimplantation may be challenging in dependent or frail patients. Moreover, incubation periods for blood cultures vary widely across laboratories and countries; guidelines suggest a minimum of 5 days [1,36], which may be extended to over 2 weeks in cases of fastidious or atypical organisms [1]. This generally contemplates a minimum waiting period of 7–8 days before reimplantation, implying an increased length of stay along with the associated costs and increased risks of acquiring nosocomial infections. Conventional practice is in line with guideline timings, with a median wait time of 3–7 days in the published series [19].

Increasing evidence suggests that the benefit–risk ratio of delaying permanent device reimplantation in a high-risk patient population is possibly limited, given a lower risk of reinfection. Findings from a systematic review and meta-analysis by Chew et al. did not support a greater risk of infection and negative outcomes in single-procedure extraction and implant. Indeed, the authors found that reimplantation beyond 72 h was associated with increased infection rates [34]. In line with their data, our observations indicate no increased risk of negative outcome and reinfection. Indeed, there were no statistical differences in outcome at 12 months. Death rates were similar across both groups, with a 12-month survival rate of 81.5% in the single-procedure and 84.6% in the SoC group, respectively (*p* = 0.075). Lost to follow-up rates were similar, with 18.5% and 15.4% in the single-procedure and SoC groups, respectively (*p* = 0.737).

In the single-procedure group, patients were followed up for 4 years, and no further deaths attributable to CIED-related infections or relapses were reported. Two patients died due to COVID-19. No other safety events were reported in the followed-up patients.

Hence, though the single procedure of extraction and reimplantation is deemed safe for patients with localized infections only and with negative pre-procedure blood cultures, our data suggest it may be feasible when applied to selected cohorts.

A similar approach has also been pursued in infected prosthetic joint infections, where a two-stage exchange is the technique of choice. While some drawbacks exist, such as peri-operative mortality, increased hospitalization length and increased healthcare costs are associated with the SoC. For all these reasons, especially in extremes of age, the single-procedure reimplant is advised in the presence of conditions that avoid relapse [37]. On the same line, a similar approach thus may be considered for some patient groups requiring device management and at high risk of negative outcomes.

Moreover, timing may also be dependent on the causative pathogen. Sohail et al. reported a median time of reimplantation of 7 days for non-bacteremic patients versus 13 days in cases of bacteremic patients. However, there was a difference in the causative organisms. Indeed, the median waiting time was 12 days in cases of *S. aureus* versus 7 days for coagulase-negative staphylococci (CoNS) [38].

Albeit not statistically assessed, the microbiology findings from both cohorts were similar and in line with those in other reported series, with coagulase-negative *Staphylococcus* spp. being the most prevalent pathogens. However, the single-procedure extraction and reimplantation cohort presented with numerically higher resistance rates and hypervirulent strains. Similar to Sohail et al., our case single-procedure series included seven *S. aureus* (42.9% methicillin-resistant S. aureus [MRSA] versus 16.7% in the SoC group) and three hypervirulent CoNS such as *S. lugdunensis*, while *S. epidermidis* also showed 61.5% resistance rates (versus 42.8% in the SoC group). In all cases, irrespective of resistance and strain, patients were re-implanted in the single-procedure group without relapsing.

In accordance with these observations, longer courses of antibiotics are generally needed in addition to device extraction when bacteremia is initially present. A previous study by Mountantonakis et al. included 15 patients without bacteremia or positive blood cultures [19], while systemic infection was reported in 18 of the 68 patients from the Nandyala et al. study, adding to the complexity of the overall cohort [18]. Just like Nandyala et al., our cohorts were also quite heterogeneous and at elevated risk of negative outcomes. In our observations, almost one-third presented with bacteremia in the single-procedure group, with a similar proportion in the SoC patient population. Moreover, three patients in the single-procedure group (11.1%) and one in the SoC group (5.1%) also presented with sepsis.

Endocarditis and concomitant bacteremia was present in eight (29.6%) and six (15.4%) of the cases in the single-procedure and SoC groups, respectively, where a positive effect of the beta-lactam combination with daptomycin/rifampicin may be more impactful [39,40,41,42].

Hence, given the current availability of novel and more active treatments with antibiofilm activity, a different approach might be worthy of consideration. Indeed, in the single-procedure reimplantation cohort, most patients underwent therapy with antibiofilm and novel agents (81.5% versus 38.5%, *p* = 0.001) with displayed potent activity on Gram-positive bacteria, while also offering coverage of difficult-to-treat and resistant pathogens, especially MRSA [22,25,26,42,43]. The optimal treatment for *S. aureus* bacteremia and/or infective endocarditis remains a subject of debate. Some in vitro studies have demonstrated the synergism between daptomycin and beta-lactams, as beta-lactam antibiotics seem to increase the sensitivity of the bacterium to daptomycin (“seesaw effect”) [44]. However, some clinical studies failed to demonstrate a positive effect of the daptomycin/beta-lactam combination therapy compared with monotherapy in terms of mortality and duration of bacteremia, particularly for MRSA bacteremia [40,41]. Further clinical studies are needed to compare daptomycin/beta-lactam combination therapy with monotherapy for CIED infections, which are frequently associated with bacterial biofilm formation, as mentioned above.

Because early and appropriate treatment is known to account for increased survival rates and decreased incidence of endocarditis complications [45], the increased availability of active molecules with biofilm penetration and a favorable pharmacokinetic/pharmacodynamic index may encourage a paradigm shift in such contexts when complexity may hinder standard-of-care approaches leading to longer times to reimplantation.

We recently reported a case of a single-procedure patient with metallo-beta-lactamase producing, meropenem-resistant *P. aeruginosa* endocarditis, who was reimplanted during imipenem plus aztreonam therapy and continued with imipenem and cefiderocol therapy. We showed that the *P. aeruginosa* strain was a biofilm producer and that the combination of imipenem and cefiderocol was synergistic against this *P. aeruginosa* in a biofilm in vitro test [46].

## 5. Conclusions

While limited to a relatively small number of patients, the data from the present study support the safety of the single-procedure extraction and reimplantation of devices, even in the presence of MDR and difficult-to-treat species or concomitant bacteremia. While extraction timing may be complex in such patients, waiting may not always be feasible. As a result, a single-procedure extraction and contralateral replacement may represent a potentially optimal solution in such instances. Moreover, the growing availability of novel molecules with greater efficacy and biofilm activity may also offer support by providing coverage and avoiding relapse.

Similar to the observations of Nandyala et al. [18], our single-procedure patient group presented with a high risk of negative outcomes and an urgent need for device reimplantation. The presence of systemic infections implied a greater risk of reinfection to a certain degree. However, aggressive antibiotic treatment may have favored early clearance and sterilization even in such a cohort, where same-day or single-procedure reimplantation may represent the strategy with the best risk/benefit balance. To this extent, the 1-year follow-up revealed no re-infections or deaths due to CIED complications.

Finally, no statistically significant differences in mortality or relapses were observed, with similar rates of patients lost to follow-up. No re-infections were reported in both cohorts, supportive of the safety and feasibility of the single-procedure approach.

The limitations of the study include its retrospective nature along with the heterogeneous antibiotic approach. In addition, the lack of randomization and clustering management in two different centers may have introduced potential bias where surgical management is concerned. Furthermore, the SoC approach referred to only the pacemaker infections in an attempt to reduce the complexity of the patients cared for in a national reference center.

To conclude, our data suggest that tailoring time to reimplantation in patients with limited alternatives and adjunctive active antibiofilm therapy may represent a novel and more feasible approach in a selected cohort of patients.

Further research is warranted in order to define the best treatment regimen and strategies to define patient selection in order to minimize the length of stay (LoS) and bed occupancy when unnecessary.

## Figures and Tables

**Figure 1 antibiotics-12-01001-f001:**
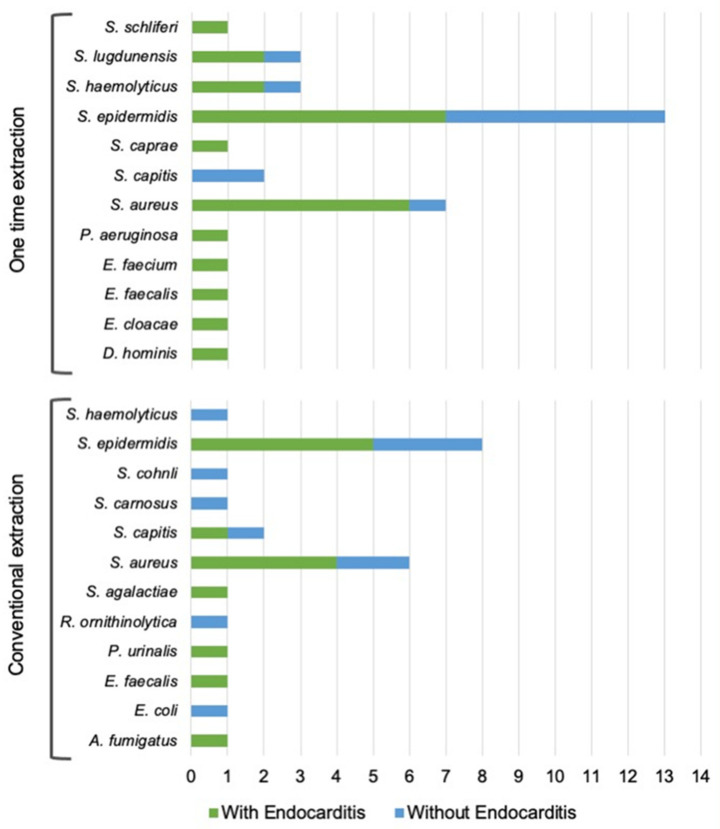
Pathogens responsible for CIED infections in both the single-procedure group and the SoC group.

**Table 1 antibiotics-12-01001-t001:** Patients characteristics in both the Single Procedure group and in the SoC group.

Patient Characteristics	Single Procedure (*n* = 27)	SoC (*n* = 39)	*p*
Sex (males)	19 (70.4%)	30 (76.9%)	0.755
Age (years)	72.07 ± 14.96	73.79 ± 8.89	0.560
Weight (kg)	73.04 ± 10.60	78.89 ± 11.51	0.046
Predisposing factors			
Skin infection	0 (0%)	4 (10.3%)	0.233
Previous endovascular infection	9 (33.3%)	5 (12.8%)	0.090
Cancer	5 (18.5%)	2 (5.1%)	0.701
Chronic kidney disease	11 (40.7%)	9 (23.1%)	0.207
Stroke	1 (3.7%)	3 (7.7%)	0.886
Dialysis	4 (14.8%)	0 (0%)	0.051
COPD	7 (25.9%)	10 (25.6%)	1
Previous CIED interventions			
Revision/upgrade/replacement/malfunction	11 (40.7%)	23 (59.0%)	0.227
Device			
Pacemaker	11 (40.7%)	39 (100%)	<0.001
Implantable cardioverter defibrillator (ICD)	6 (22.2%)	0 (0%)	0.008
Biventricular defibrillator implant	10 (37.0%)	0 (0%)	<0.001
Device extraction and implant			
Time from device implant (years)	2 (1–7)	9 (4–13)	0.002
Time from diagnosis to extraction (days)	27 (12–30)	60 (30–90)	<0.001
Length of procedure (minutes)	140 (102–180)	151 (120–224)	0.192
Final diagnosis			
Endocarditis	18 (66.7%)	17 (43.06%)	0.082
Localized infection	17 (33.3%)	18 (56.4%)	0.144
Vegetation			
Vegetation	14 (51.9%)	15 (38.5%)	0.409
Vegetation size (median, range-mm)	3.12 ± 0.91	4.07 ± 1.12	0.857
Positive blood culture	12 (44.44%)	13 (33.3%)	0.511
Concomitant sepsis	3 (11.1%)	2 (5.1%)	0.393
Anticoagulant therapy	15 (55.6%)	16 (41.0%)	0.318
Antibiotic therapy (admission)			
Overall	9 (33.3%)	21 (53.6%)	0.163
Anti-Gram-negative *	7 (25.9%)	9 (23.0%)	1.000
Antibiotic therapy (hospitalization)			
Overall	24 (88.9%)	39 (100%)	0.064
Anti-Gram-negative *	20 (74.1%)	15 (38.5%)	0.006
Activity against biofilm ^†^	22 (81.5%)	15 (38.5%)	0.001
Outcome			
Survival (1 month)	24 (88.9%)	39 (100%)	0.126
Survival (12 months)	22 (81.5%)	33 (84.6%)	0.737
Lost to follow-up	5 (18.5%)	6 (15.4%)	0.737
Reinfection	0 (0%)	NA	---

* Anti-Gram-negative: antibiotic agents with displayed activity against Gram-negative bacteria. ^†^ Activity against biofilm: antibiotic combination (daptomycin/rifampicin + beta-lactam) with antimicrobial activity demonstrated in vitro against common biofilm-producing bacteria isolated in our cohort [22,23,24].

**Table 2 antibiotics-12-01001-t002:** Antibiotic treatment during hospitalization.

Antibiotic	**Single Procedure** **(27)**	**SoC** **(39)**
Overall	24 (88.9%)	39 (100%)
Oxacillin/flucloxacillin	0 (0%)	4 (10.3%)
Cefazolin	7 (25.9%)	0 (0%)
Daptomycin	20 (74.1%)	20 (51.3%)
Vancomycin	0 (0%)	2 (5.13%)
Teicoplanin	0 (0%)	5 (12.8%)
Linezolid	0 (0%)	2 (5.13%)
Ceftobiprole	7 (25.9%)	0 (0%)
Ceftaroline	2 (7.41%)	3 (7.69%)
Others	5 (18.5%)	20 (51.3%)
Monotherapy	3 (11.1%)	14 (35.9%)
Combination therapy	21 (77.8%)	25 (64.1%)

## Data Availability

The data presented in this study are available on request from the corresponding author. The data are not publicly available due to privacy reasons.

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
