# Peer review of "Safety and Efficacy of a Single Procedure of Extraction and Reimplantation of Infected Cardiovascular Implantable Electronic Device (CIED) in Comparison with Deferral Timing: An Observational Retrospective Multicentric Study"

_antibiotics, 2023, doi:10.3390/antibiotics12061001_

Round 1
Reviewer 1 Report
“Safety and efficacy of a single procedure of extraction and re-implantation in comparison with deferral timing: an observational retrospective multicentric study. “ is the retrospective study to compare the outcome between two treatment method for CIED extraction and reimplantation. Clinicians are straggling to decide reimplantation timing for CIED infection. This study may have some good information for cardiologists and cardiovascular surgeon.
Here show some comments.
1. P2 L64, “Patients with local, systemic in-64 fection or infective endocarditis CIED-related were enrolled.” The author didn’t include endocarditis patients. Please delete this sentence or rewrite.
2. Two hospitals are study site. However, each hospital has different treatment strategy. This is the huge bias for the result of this study.
3. This study didn’t include endocarditis patient. It means that this study doesn’t include severe patients.
4. As mentioned above, this is a retrospective study with very large and unreliable bias in the number of patients, patient selection, and treatment strategies.
5. However, it may be useful information in that it shows the possibility of treating a pacemaker infection in a single stage.
This manuscript is well written and only minor corrections are needed regarding the English language.
Author Response
- P2 L64, “Patients with local, systemic in-64 fection or infective endocarditis CIED-related were enrolled.” The author didn’t include endocarditis patients. Please delete this sentence or rewrite.
A: Dear referee, in our cohort there were 18 (66.7%) Infective endocarditis in the single-procedure group and 17 (43%) in the SoC group, as shown in the Table 1
- Two hospitals are study site. However, each hospital has different treatment strategy. This is the huge bias for the result of this study.
A: Dear referee, thank you for your comment, we know very well the bias of our study. This an exploratory observational retrospective study to assess the efficacy of single-procedure in order to design future randomized prospective study. This observation are included among the limitation of this study
- This study didn’t include endocarditis patient. It means that this study doesn’t include severe patients.
A: Dear referee, in our cohort there were 18 (66.7%) Infective endocarditis in the single-procedure group and 17 (43%) in the SoC group, as shown in the Table 1. Furthermore, at admission 12 patients (44%) in the single procedure and 13 (33%) in the SoC group had positive blood cultures, 3 patients in the single-procedure group and 2 patients in the SoC group had concomitant sepsis (table 1). According to these data severe patients or patients with uncontrolled infections were enrolled in both arms of the study.
- As mentioned above, this is a retrospective study with very large and unreliable bias in the number of patients, patient selection, and treatment strategies.
A: As stated before, we are aware of limitations of this study, but again this is only a pilot study. Limitations are highlighted in the conclusions section
- However, it may be useful information in that it shows the possibility of treating a pacemaker infection in a single stage.
A: Thank you very much for your comments and constructive feed-back
Reviewer 2 Report
“Safety and efficacy of a single procedure of extraction and re-implantation in comparison with deferral timing: an observational retrospective multicentric study.” By Tascini et al. The paper is a study and comparison of the outcome and safety of single-procedure and two-stage replacement procedure of infected cardiovascular implantable electronic device implantations. The study is on a significant medical problem and presents interesting results. However, the authors need to pay more attention to presenting the results more clearly and make the language more understandable. I am pointing out some of these in the following comments:
Line 23: “side-to-side device extraction” Not clear what this means. Is this a medical terminology? Does “side-to-side” refer to comparison of two procedures?
Line 44: Change “infections rate account” to “infections account” or “infection rates account”
Line 71: “we have decided to enroll only patients with pace-maker infections that normally have less severe infections compared to indwelling defibrillators.” The verb does not match the subject. Please change the grammar.
Line 76: “In the single procedure cohort, the small theoretical risk for reinfection of the new device was felt to be out-weighed by the benefit of early reimplantation of a contralateral device in the setting of device-dependence.” Please make the language more simple and easier to understand.
Line 95: “Instead” A paragraph cannot start with “Instead”
Line 97: “laser energy of 308 nm” nm is not a unit of energy.
Line 115-187: Not clear why the results are presented in so many paragraphs. There are 19 paragraphs in these two pages with one to two sentences per paragraph.
Line 115-187: There is no need to repeat all information that one can easily see in the table and figure, especially all the numbers and statistical analyses. Repeating so much of numerical information is unnecessary, monotonous and confusing. Please state the important findings, the reader can read the rest of the data from the table and figure.
Line 134: “SOC” written as “SoC” in the rest of the manuscript.
Line 134: “Positive blood cultures” I guess, this means positive for (bacterial) infection. Need to state that.
Line 137: “no significance in differences” I guess, the authors mean “no significant difference”
Line 160 and 166: “Figures 1 and 2”. But there is only one figure in the manuscript.
Line 170 and 178: “Anti-Gram-negative coverage” Not clear what this means. Please clarify.
Line 174: “active anti-biofilm antimicrobial therapy (including daptomycin or rifampicin alone or in combination with a beta lactam, especially novel 5th generation molecules) Not clear what “including means here. Are the antibiotics mentioned the only ones used or were there many antibiotics including these ones named? Also, the meaning of “anti-biofilm antimicrobial” is not clear. Are these antibiotics anti-biofilm or antimicrobial or both or active against bacteria that form biofilms? The antimicrobial activity of these may not need a reference, but if they have anti-biofilm activity, please provide reference(s). Please provide a complete list of all antibiotics that the patients were using and which ones of those are classified as anti-Gram negative, anti-biofilm etc.
Line 188 (Table 1): “Antibiotic Therapy (admission)” and “Antibiotic Therapy (hospitalization)” Most people will not understand what these terms in parentheses mean. Please state more clearly. I guess, the authors are trying to say (before admission) and (after admission).
Line 182: “No serious adverse events occurred within the first month and thereafter.” Not clear what the authors are trying to say. The sentence as written, means that there was no adverse effect at all throughout the process.
Line 190: Figure 1: According to the literature, it is widely accepted that the HACEK (Haemophilus species, Aggregatibacter actinomycetemcomitans, Cardiobacterium hominis, Eikenella corrodens, and Kingella kingae) group of bacteria are usually associated with endocarditis. However, the species listed in Figure 1 do not agree with this list of HACEK bacteria. Authors please comment on this.
Line 281 “pk/pd” While this may be a common terminology in medicine and pharmacy, most readers may not be familiar with this. Please write the full form.
Line 284: “metallo-enzyme” If the authors are referring to b-lactamase, please state that, because all readers may not know that.
The authors need to pay more attention to presenting the results more clearly and make the language more understandable.
Author Response
Line 23: “side-to-side device extraction” Not clear what this means. Is this a medical terminology? Does “side-to-side” refer to comparison of two procedures?
A: we have modified “side-to-side device extraction” with “device extraction and contralateral reimplantation”
Line 44: Change “infections rate account” to “infections account” or “infection rates account”
A: thank you for your advise. We have changed it accordingly
Line 71: “we have decided to enroll only patients with pace-maker infections that normally have less severe infections compared to indwelling defibrillators.” The verb does not match the subject. Please change the grammar.
A: thank you for your advise. We have changed it accordingly
Line 76: “In the single procedure cohort, the small theoretical risk for reinfection of the new device was felt to be out-weighed by the benefit of early reimplantation of a contralateral device in the setting of device-dependence.” Please make the language more simple and easier to understand.
A: We have modified this phrase with “In the single procedure cohort, the benefit of early reimplantation of a contralateral device in the setting of device-dependence was considered to outweigh the small theoretical risk for reinfection of the new device.”
Line 95: “Instead” A paragraph cannot start with “Instead”
A: thank you for your advise. We have changed it accordingly
Line 97: “laser energy of 308 nm” nm is not a unit of energy.
A: We have changed it with “cool” pulsed ultraviolet laser at a wavelength of 308 nm”
Line 115-187: Not clear why the results are presented in so many paragraphs. There are 19 paragraphs in these two pages with one to two sentences per paragraph.
A: The text was amended accordingly
Line 115-187: There is no need to repeat all information that one can easily see in the table and figure, especially all the numbers and statistical analyses. Repeating so much of numerical information is unnecessary, monotonous and confusing. Please state the important findings, the reader can read the rest of the data from the table and figure.
A: thank you for your advise. We have changed it accordingly
Line 134: “SOC” written as “SoC” in the rest of the manuscript.
A: thank you for your advise. We have changed it accordingly
Line 134: “Positive blood cultures” I guess, this means positive for (bacterial) infection. Need to state that.
A: thank you for your advise. We have changed it with “Positive blood culture for bacteria at the admission to the hospital”
Line 137: “no significance in differences” I guess, the authors mean “no significant difference”
A: thank you for your advise. We have changed it accordingly
Line 160 and 166: “Figures 1 and 2”. But there is only one figure in the manuscript.
A: thank you for your advise. We have changed it accordingly
Line 170 and 178: “Anti-Gram-negative coverage” Not clear what this means. Please clarify.
A: We have modified it with “Antibiotic therapy active against Gram-negative bacteria” and we have added a footer at Table 1 where we clarify what we mean with “anti-Gram-negative”
Line 174: “active anti-biofilm antimicrobial therapy (including daptomycin or rifampicin alone or in combination with a beta lactam, especially novel 5th generation molecules) Not clear what “including means here. Are the antibiotics mentioned the only ones used or were there many antibiotics including these ones named? Also, the meaning of “anti-biofilm antimicrobial” is not clear. Are these antibiotics anti-biofilm or antimicrobial or both or active against bacteria that form biofilms? The antimicrobial activity of these may not need a reference, but if they have anti-biofilm activity, please provide reference(s). Please provide a complete list of all antibiotics that the patients were using and which ones of those are classified as anti-Gram negative, anti-biofilm etc.
A: The daptomycin or rifampicin alone or in combination with a beta lactam was the only anti-biofilm therapy used in our cohort. We have added a Definitions paragraph where we clarify this point and what we mean with “anti-Gram-negative” and “anti-biofilm activity”, providing some references about it. We have added a table to list the antibiotics most used in our cohort
Line 188 (Table 1): “Antibiotic Therapy (admission)” and “Antibiotic Therapy (hospitalization)” Most people will not understand what these terms in parentheses mean. Please state more clearly. I guess, the authors are trying to say (before admission) and (after admission).
A: We have changed them with “at admission” and “during hospitalization” respectively
Line 182: “No serious adverse events occurred within the first month and thereafter.” Not clear what the authors are trying to say. The sentence as written, means that there was no adverse effect at all throughout the process.
A: We mean that there was no serious adverse effect related to the procedure or the antibiotic therapy at all throughout the process. We have removed the time space
Line 190: Figure 1: According to the literature, it is widely accepted that the HACEK (Haemophilus species, Aggregatibacter actinomycetemcomitans, Cardiobacterium hominis, Eikenella corrodens, and Kingella kingae) group of bacteria are usually associated with endocarditis. However, the species listed in Figure 1 do not agree with this list of HACEK bacteria. Authors please comment on this.
A: The figure 1 illustrates only the microorganisms responsible for the CIED infections of our cohort, not the pathogens most commonly associated with endocarditis according to the literature. We have specified it in the footer of the Figure 1
Line 281 “pk/pd” While this may be a common terminology in medicine and pharmacy, most readers may not be familiar with this. Please write the full form.
A: We have changed it accordingly
Line 284: “metallo-enzyme” If the authors are referring to b-lactamase, please state that, because all readers may not know that.
A: We have changed it accordingly
Reviewer 3 Report
The paper compares safety and efficacy between a single procedure of extraction and re-implantation and SOC in patients with CIED infections. It is an interesting topic. I have the following comments:
I suggest that the title include some words about CIED infection. All readers may not understand what the paper is all about with the present title.
The patients from the single procedure centre, were they treated with there regular guidelines for CIED extractions or were they exceptions from the guidelines due to severity of there cardiac illness?
Materials and Methods:
2.2 R 105 Did all patients receive antibiotic prophylaxis (and which antibiotics?)
2.3 Microbiology. It is described that the microbiology was cultures from leads or infected material. Not blood cultures? It is described in the result part that some patients had positive blood cultures. 16 S rRNA from leads was not used in these patients?
Results:
R131-138. In this parts the authors present results about local infection, endocarditis, positive blood cultures, endocarditis + positive blood cultures. The presentation is unclear. One criteria for endocarditis is positive blood culture. Maybe this could be better presented in a table
R 139-141. Vegetations is mentioned here. Was TEE performed, what were the locations of the vegetations, leads? valves?
R 150-153 Time from diagnosis of CIED infection until extraction seems to be very long 27 and 60 days respectively, is this common?
R 157-167 The microbiological results are difficult to follow in the text. I suggest a table. Are these cultures from leads only? What about blood cultures? I cannot find figure 2.
R 176. Patients in the single procedure group had a higher proportion of antibiotics with biofilm activity than the SOC group, they were treated wot daptomycin, rifampicin. Do you have about data about the length of this biofilm antibiotic treatment? Patients in the single procedure group seem to have a higher degree of methicillin resistant staphylococci, did that explain the choice of antibiotics?
Table 1
Final diagnosis, what is the difference between endocarditis and endocarditis + BSI?
It is acceptable
Author Response
I suggest that the title include some words about CIED infection. All readers may not understand what the paper is all about with the present title.
A: We have changed it accordingly
The patients from the single procedure centre, were they treated with there regular guidelines for CIED extractions or were they exceptions from the guidelines due to severity of there cardiac illness?
A: The patients in the single procedure group were treated according to the European guidelines for the treatment of CIED infections, except for the timing of the reimplantation
Materials and Methods:
2.2 R 105 Did all patients receive antibiotic prophylaxis (and which antibiotics?)
A: The patients were all under antibiotic treatment at the time of extraction and reimplantation of the CIED, as they suffered from an infection
2.3 Microbiology. It is described that the microbiology was cultures from leads or infected material. Not blood cultures? It is described in the result part that some patients had positive blood cultures. 16 S rRNA from leads was not used in these patients?
A: The microorganisms responsible for the CIED infections were isolated both from leads cultures and/or blood cultures. No we did no use the 16 S rRNA technique
Results:
R131-138. In this parts the authors present results about local infection, endocarditis, positive blood cultures, endocarditis + positive blood cultures. The presentation is unclear. One criteria for endocarditis is positive blood culture. Maybe this could be better presented in a table
A: We have removed the voice endocarditis + positive blood cultures. The absolute numbers and the percentages of the endocarditis, the local infections and the positive blood culture are presented in Table 1 yet
R 139-141. Vegetations is mentioned here. Was TEE performed, what were the locations of the vegetations, leads? valves?
A: Yes all the patients underwent TEE. We have clarified what we mean with vegetations in the Definitions paragraph. Unfortunately, we do not have data relative to the location of the vegetations
R 150-153 Time from diagnosis of CIED infection until extraction seems to be very long 27 and 60 days respectively, is this common?
A: Yes, because the Hospital of Pisa is a referral center for the CIED infections in Italy and many patients are referred to that centre from different parts of the country
R 157-167 The microbiological results are difficult to follow in the text. I suggest a table. Are these cultures from leads only? What about blood cultures? I cannot find figure 2.
A: The pathogens responsible for CIED infections in our cohort are presented in figure 1. In the Definitions paragraph we have specified that the microorganisms were isolated from both lead cultures and/or blood cultures
R 176. Patients in the single procedure group had a higher proportion of antibiotics with biofilm activity than the SOC group, they were treated with daptomycin, rifampicin. Do you have about data about the length of this biofilm antibiotic treatment? Patients in the single procedure group seem to have a higher degree of methicillin resistant staphylococci, did that explain the choice of antibiotics?
A: No, the anti-biofilm activity is not related to methicillin resistance. In the single procedure cohort anti-biofilm treatments have been used more frequently in order to reduce infection risk, since in this group of patients the reimplanted material might be colonized by the causative pathogen.
Table 1
Final diagnosis, what is the difference between endocarditis and endocarditis + BSI?
A: We have removed the voice endocarditis + positive blood cultures.
Round 2
Reviewer 1 Report
The author haven't revised in each comment. I recommend the author that the author has to reply each reviewers comments. Please read reviewers comments again and revise your manuscript.
Author Response
“Safety and efficacy of a single procedure of extraction and re-implantation in comparison with deferral timing: an observational retrospective multicentric study. “ is the retrospective study to compare the outcome between two treatment method for CIED extraction and reimplantation. Clinicians are straggling to decide reimplantation timing for CIED infection. This study may have some good information for cardiologists and cardiovascular surgeon.
Here show some comments.
- P2 L64, “Patients with local, systemic in-64 fection or infective endocarditis CIED-related were enrolled.” The author didn’t include endocarditis patients. Please delete this sentence or rewrite.
A: Dear referee, in our cohort there were 18 (66.7%) Infective endocarditis in the single-procedure group and 17 (43%) in the SoC group, as shown in the Table 1
- Two hospitals are study site. However, each hospital has different treatment strategy. This is the huge bias for the result of this study.
A: Dear referee, thank you for your comment, we know very well the bias of our study. This an exploratory observational retrospective study to assess the efficacy of single-procedure in order to design future randomized prospective study. This observation are included among the limitation of this study
- This study didn’t include endocarditis patient. It means that this study doesn’t include severe patients.
A: Dear referee, in our cohort there were 18 (66.7%) Infective endocarditis in the single-procedure group and 17 (43%) in the SoC group, as shown in the Table 1. Furthermore, at admission 12 patients (44%) in the single procedure and 13 (33%) in the SoC group had positive blood cultures, 3 patients in the single-procedure group and 2 patients in the SoC group had concomitant sepsis (table 1). According to these data severe patients or patients with uncontrolled infections were enrolled in both arms of the study.
- As mentioned above, this is a retrospective study with very large and unreliable bias in the number of patients, patient selection, and treatment strategies.
A: As stated before, we are aware of limitations of this study, but again this is only a pilot study. Limitations are highlighted in the conclusions section
- However, it may be useful information in that it shows the possibility of treating a pacemaker infection in a single stage.
A: Thank you very much for your comments and constructive feed-back
